# PiML: Automated Machine Learning Workflow Optimization using LLM Agents

**Abhishek Chopde**[1,*] **Fardeen Pettiwala**[1,*] **Kirubananth Sankar**[1,*] **Sai Botla**[1,*]
**Pachipulusu Ayyappa Kethan**[1]

[1]Fractal AI Research, Mumbai.
[*]Equal contribution.

**Abstract** In this paper, we introduce *PiML-Persistent Interative Machine Learning* agentic framework, a novel automated pipeline specifically designed for solving real-world machine learning (ML) tasks such as Kaggle competions. PiML integrates iterative reasoning, automated code generation, adaptive memory construction, and systematic debugging to tackle complex problems effectively. To rigorously assess our framework, we selected 26 diverse competitions from the MLE-Bench benchmark, ensuring comprehensive representation across various complexity levels, modalities, competition types, and dataset sizes. We quantitatively compared PiML's performance to AIDE—the best-performing existing baseline from MLE-Bench—across multiple evaluation metrics: Valid Submission rate, Submissions Above Median, Average Percentile Rank, and Medal Achievement Rate. Using the "o3-mini" model, PiML surpassed the baseline in submissions above median (41.0% vs 30.8%), medal attainment rate (29.5% vs 23.1%), and average percentile rank (44.7% vs 38.8%). These results highlight PiML's flexibility, robustness, and superior performance on practical and complex ML challenges.

## 1 Introduction

Designing an end-to-end machine learning (ML) workflow is a complex effort that requires substantial expertise, as manually crafting and optimizing these workflows for specific tasks is both labor-intensive and knowledge-intensive. This challenge has been partially addressed by AutoML (Erickson et al., 2020a; Tang et al., 2024; Shchur et al., 2023), which automates various stages of the workflow, streamlining processes that would otherwise demand extensive human effort (Feurer et al., 2015). However, while AutoML has improved efficiency to a degree, it operates within a predefined rule set and often lacks the flexibility necessary to adapt to the domain specific requirements of the problem (Zöller and Huber, 2021).

In contrast, the emergence of Large Language Models (LLMs) has revolutionized problem-solving approaches thanks to their expansive knowledge bases and advanced reasoning capabilities. Techniques like Chain of Thought (CoT)(Wei et al., 2022), Tree of Thoughts (ToT)(Yao et al., 2023a), and ReAct(Yao et al., 2023b) have demonstrated the potential of LLMs in tackling complex coding tasks, showcasing their ability to facilitate complex reasoning processes. These capabilities can be applied to ML workflows, offering potential solutions to previously challenging downstream tasks. Many works attempted to address some parts of a ML workflow - Feature Engineering (Hollmann et al., 2023; Jeong et al., 2024; Zhang et al., 2024b; Gong et al., 2024; Li et al., 2025; Malberg et al., 2024), Model Selection – (Zhang et al., 2023; Shen et al., 2023), HPO – (Liu et al., 2025; Zhang et al., 2024a).

We propose a novel multi-agent framework, *PiML: **Persistent Interative Machine Learning** agent* for exploring the true exploratory nature of ML problem solving via iterative experimentation. Unlike many other similar works, our framework enables a dynamic step-by-step approach to problem solving.

Our contributions include:

1. **Automated Agent Pipeline:** We introduce PiML, a structured and iterative automated pipeline that systematically leverages an agent's internal reasoning ("Thoughts") and executable code ("Actions") guided by summarized execution feedback ("Observations") to solve machine learning tasks from Kaggle competitions autonomously.

2. **Adaptive Memory Management**: A novel multi-tier memory construction strategy, effectively balancing detailed recent context with summarized historical actions.

3. **Robust Error Handling via Debug Chain**: We present a structured and systematic "Debug Chain" mechanism that iteratively refines erroneous code actions generated by the pipeline, improving error resolution and enabling efficient self-correction without human supervision.

4. **Experimental Validation and Competitiveness**: Empirical evaluation on the diverse MLE-Bench (MLE-Pi - Our statistically sampled dataset 4.3) dataset demonstrates the flexibility and effectiveness of PiML. Specifically, our results indicate superior or competitive performance against strong baseline automated frameworks, thereby highlighting PiML's potential to achieve competitive results across various competition complexities and categories autonomously.

## 2 Related Work

Large language models (LLMs) have rapidly progressed from System-1 architectures that rely on pattern recognition (OpenAI, 2024; Anthropic, 2024) to System-2 variants that explicitly plan and reason (OpenAI, 2025b,c; DeepSeek-AI, 2025; Anthropic, 2025). This evolution has unlocked a broad spectrum of agentic applications, including autonomous code generation (Le et al., 2022; Singh et al., 2025) and debugging (Chen et al., 2023; Zhong et al., 2024), complex decision making in finance, healthcare, and patient care (Peng et al., 2023; Busch et al., 2025), and research automation (Gottweis et al., 2025; Lu et al., 2024). For machine-learning workflows, these reasoning-centric models have made end-to-end AutoML more attainable: systems must coordinate data exploration, feature engineering, model selection, and hyperparameter optimization (HPO) holistically and iteratively. Traditional AutoML toolkits such as AutoGluon, H2O, and Auto-Sklearn (Erickson et al., 2020b; LeDell et al., 2020; Feurer et al., 2015) still rely on fixed heuristics and treat each sub-problem independently, which often yields sub-optimal results.

Building on this foundation, several agentic frameworks now tackle long-horizon tasks. Open-Hands (Wang et al., 2024) automates software development tasks by combining code interaction, execution, and web search, while OpenManus (Liang et al., 2025; manus.im, 2025) generalizes this strategy to broader multi-step problems. In data science, AIDE (Jiang et al., 2025) employs a tree-structured search that incrementally explores alternative solution paths; AutoKaggle(Li et al., 2024) orchestrates five specialized agents—Reader, Planner, Developer, Reviewer, and Summarizer—to cover the full ML pipeline; DS-Agent (Guo et al., 2024) blends LLMs with case-based reasoning to leverage historical Kaggle solutions, and Agent-K (Grosnit et al., 2024) adds nested memory processing to support continuous improvement. Collectively, these systems demonstrate a clear trend towards autonomous agents that reduce human intervention while elevating the efficiency and quality of complex, multi-stage workflows.

## 3 Methodology

In this section, we describe our automated agent pipeline designed to solve machine learning tasks such as Kaggle competitions.

### 3.1 Framework Overview

PiML is an end-to-end framework for machine learning, as illustrated in Figure 1 and Algorithm 1. The system accepts a Task Description as input and autonomously develops and refines machine learning solutions. The Task Description (for example, in the context of a Kaggle competition)

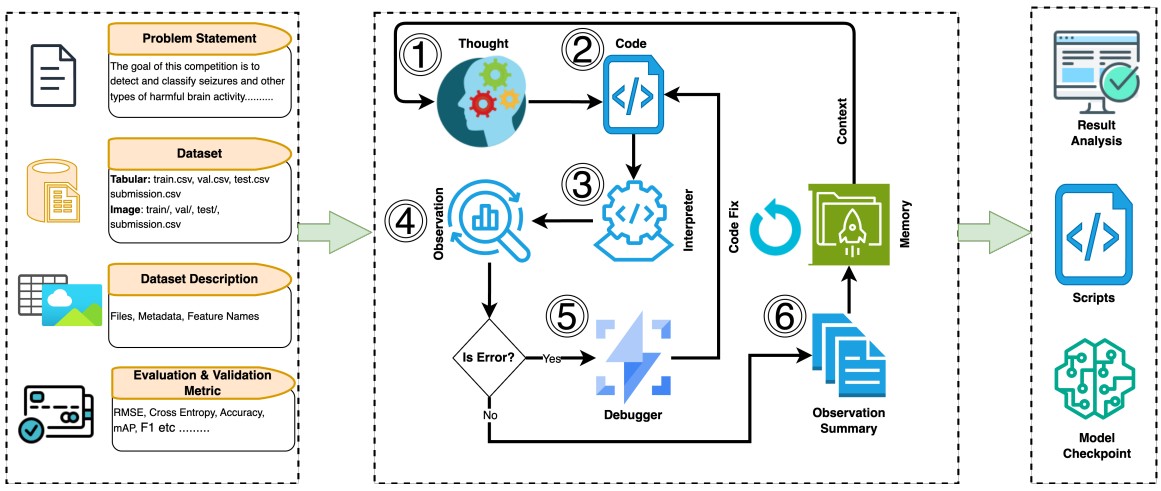

Figure 1: PiML is an end-to-end framework for autonomous machine learning. Given the problem description, dataset, and evaluation metric, the framework can iteratively perform EDA, Feature Engineering, modeling, and hyperparameter tuning to obtain the best results.

typically includes information about the competition objectives, evaluation criteria, rules, required constraints, and an initial overview of the dataset.

At its core, PiML operates through an iterative thought-action-observation cycle orchestrated by specialized agents. The Main Agent is responsible for generating both the reasoning process (thought) and the corresponding code implementation (action). The Summary Agent generates feedback summaries to guide the next cycle. The Debug chain mechanism identifies and resolves errors encountered during execution, while the Memory mechanism maintains contextual information across cycles to support continual learning and adaptation. Throughout the execution, PiML produces submission files representing complete solution attempts to the given task.

These cycles continue until reaching predefined constraints, systematically exploring the solution space to optimize performance on the specified task. The following sections detail each component of this process and its interactions within the framework.

### 3.2 Iterative Refinement Cycle

We formally define a single cycle of our pipeline, which we refer to as a Step, as:

$$S = (T, A, O, \hat{O}),$$

where $T$ denotes a Thought and $A$ denotes an Action (i.e., the generated code) generated by the Main Agent at step $S$, $O$ denotes the Observation resulting from the execution of $A$, and $\hat{O}$ denotes the summary of the observation.

Each step $S$ represents a single thought-action-observation cycle in our pipeline. The creation of a step is mainly influenced by a fixed Task Description $\mathcal{D}$, which remains unchanged throughout execution, and the Memory of all preceding steps, denoted by $\mathcal{M}$.

We define the agents and functions used in each step. The Main Agent, represented as *MainAgent*, is responsible for generating a Thought and an Action based on the given input task and memory. The Code Interpreter, represented as *ExecuteCode*, executes an Action and generates the corresponding Observation. The Summary Agent, represented as *SummaryAgent*, analyzes an observation and produces the Observation Summary. The debug chain mechanism represented as *DebugChain*, assists in identifying and resolving errors encountered during the execution of an Action. The sequence of all steps generated up to the current point forms the agent

trajectory, denoted as $\tau = \{S_1, S_2, \ldots, S_i\}$, where $i$ is the current step index. Finally, the Memory Constructor, represented as $ConstructMemory$, formulates the memory from the agent trajectory.

The sequence of operations within each step is as follows:

1. **Thought and Action Generation**: At the start of each step, the Main Agent produces a thought and action based on the task description ($\mathcal{D}$) and memory ($\mathcal{M}$).
$$(T, A) = MainAgent(\mathcal{D}, \mathcal{M})$$

2. **Execution and Observation**: The action (generated code) is then executed to produce an Observation.
$$O = ExecuteCode(A)$$

3. **Observation Summary**: The observation is then analyzed and summarized by Summary Agent.
$$\hat{O} = SummaryAgent(O, A)$$

4. **Debug** (if Needed): If errors are detected, a debugging chain is initiated with a maximum allowed debug steps $DS$.

$$(T, A, O, \hat{O}) = DebugChain(\mathcal{D}, T, A, O, \hat{O}, DS)$$

5. **Update Trajectory**: The current step is added to the agent trajectory.
$$S = \{(T, A, O, \hat{O})\}$$
$$\tau = \tau \cup S$$

6. **Memory Construction**: At the end of each iteration, memory is constructed for the next step based on the updated trajectory.
$$\mathcal{M} = ConstructMemory(\tau)$$

## 3.3 Result Generation and Optimization

During pipeline execution, the Main Agent autonomously generates submission files, conditioned on its internal reasoning and the feedback acquired through observations. Each submission file constitutes a complete solution attempt for the specified task. The collection of all such submission files forms the final result set.

Formally, we define the set of submissions created by our pipeline as:

$$\mathcal{R} = \{R_1, R_2, \ldots, R_N\}$$

where each $R_j$, for $j \in \{1, 2, \ldots, N\}$, represents a submission file generated during some step of execution. As shown in Algorithm 1, whenever an action $A$ produces a valid submission, it is added to the submission set $\mathcal{R}$.

Through iterative generation of multiple submissions, our pipeline progressively explores diverse solution approaches while leveraging intermediate feedback. This iterative refinement mechanism facilitates optimization within the predefined execution constraints of time ($\bar{T}$) and total steps ($TS$), maximizing the likelihood of achieving superior outcomes.

## 3.4 Memory Construction Dynamics

The function call $ConstructMemory(\tau, L)$ in Algorithm 1 systematically constructs memory $\mathcal{M}$ from the trajectory $\tau$ while adhering to token-length limitations ($L$). This construction is crucial for providing the Main Agent with sufficient context for generating informed thoughts and actions in subsequent steps.

---

**Algorithm 1** PiML: Overall Algorithm

---

**Input**: $\mathcal{D}$: task description, $\bar{T}$: time limit, $TS$: total steps limit, $DS$: debug steps limit, $L$ : token length limit
**Output**: submission set $\mathcal{R}$

Initialize trajectory $\tau \leftarrow \emptyset$, memory $\mathcal{M} \leftarrow \epsilon$, submissions $\mathcal{R} \leftarrow \emptyset$
Initialize step counter $i \leftarrow 0$, time $t \leftarrow 0$
**while** $t < \bar{T}$ **and** $i < TS$ **do**
    ▶ *Generate thought and action*
    $(T, A) \leftarrow MainAgent(\mathcal{D}, \mathcal{M})$
    ▶ *Execute and observe*
    **if** $A$ is valid **then**
        $O \leftarrow ExecuteCode(A)$
    **else**
        $O \leftarrow O_{\text{default}}$
    **end if**
    $\hat{O} \leftarrow SummaryAgent(O, A)$
    ▶ *Debug if error detected*
    **if** $\hat{O}$ indicates error **then**
        $(T, A, O, \hat{O}) \leftarrow DebugChain(T, A, O, \hat{O}, \mathcal{D}, DS)$
    **end if**
    ▶ *Construct step*
    $S = (T, A, O, \hat{O})$
    ▶ *Collect submission*
    **if** $A$ generates submission $R_{\text{current}}$ **then**
        $\mathcal{R} \leftarrow \mathcal{R} \cup \{R_{\text{current}}\}$
    **end if**
    ▶ *Update trajectory, memory and system state*
    $\tau \leftarrow \tau \cup S$
    $\mathcal{M} \leftarrow ConstructMemory(\tau, L)$
    $i \leftarrow i + 1, t \leftarrow t + \Delta t$
**end while**
**return** $\mathcal{R}$

---

Let the trajectory up to the current step $i$ be:
$$\tau = \{S_1, S_2, \ldots, S_i\}, \quad \text{where} \quad S_i = (T_i, A_i, O_i, \hat{O}_i)$$
The memory $\mathcal{M}$ is constructed using one of the following prioritized strategies, selected based on whether the constructed memory fits within the token constraint $L$:

**Strategy 1: Comprehensive Memory**:
    Includes full context (thoughts, actions, and summarized observations) for the most recent $w$ steps, and actions from earlier steps:
$$\mathcal{M} = \{A_k \mid 1 \le k \le i - w\} \cup \{(T_j, A_j, \hat{O}_j) \mid i - w + 1 \le j \le i - 1\} \cup \{(T_i, A_i, O_i)\}$$

**Strategy 2: Reduced Recent Context**:
    If Strategy 1 exceeds $L$, retain all past actions and only the current full step:
$$\mathcal{M} = \{A_k \mid 1 \le k \le i - 1\} \cup \{(T_i, A_i, O_i)\}$$

**Strategy 3: Historical Action Chain:**
> If Strategy 2 exceeds $L$, include as many previous actions as possible, starting from step $m$, where $m$ is the largest index satisfying the constraint:
> $$\mathcal{M} = \{A_k \mid m \leq k \leq i - 1\} \cup \{(T_i, A_i, O_i)\}, \quad 1 \leq m \leq i$$

**Strategy 4: Current Step Only:**
> If Strategy 3 exceeds $L$, include only the current step:
> $$\mathcal{M} = \{(T_i, A_i, O_i)\}$$

**Strategy 5: Minimal Context:**
> The fallback when only a minimal signal can be encoded:
> $$\mathcal{M} = \{(A_i, O_i)\}$$

The threshold of $w = 10$ recent steps in Strategy 1 represents an empirically determined balance between comprehensive context and computational efficiency. This threshold provides sufficient recent problem-solving history while preventing memory overload. Our experimental observations indicate that maintaining full context for approximately 10 steps captures the most relevant recent problem-solving decisions while allowing room for longer-term action history that establishes the solution trajectory. This hybrid approach ensures the agent maintains both detailed recent context and broader historical perspective.

At each memory construction stage, the *ConstructMemory* function sequentially evaluates each strategy until finding one that satisfies the token constraints. This adaptive approach optimizes the information provided to the Main Agent within system limitations.

### 3.5 Debug Chain: Systematic Error Correction

When an error is detected in the observation summary $\hat{O}$ of the current step, the debug chain mechanism is activated. As shown in Algorithm 1, the function $DebugChain(T, A, O, \hat{O}, \mathcal{D}, DS)$ systematically attempts to resolve the error through an iterative refinement process. The debug chain iteratively refines the action $A$ until either the error is successfully resolved or the maximum debug steps limit $DS$ is reached.

The mechanism works by diagnosing errors, generating improved versions of the action, executing these refinements, and evaluating their outcomes. Even when errors persist after reaching the maximum debug depth, the exploration provides valuable context for the Main Agent's subsequent reasoning.

The Debug Chain function returns the updated tuple $(T, A, O, \hat{O})$ where $T$ represents the final thought that integrates insights from all debug iterations, $A$ represents the final action, either the successfully corrected version or the last attempted refinement, $O$ represents the observation from executing the final action and $\hat{O}$ represents the summary of the final observation.

This mechanism enables the agent to recover from execution errors within a single step rather than requiring multiple main steps for error correction, thereby enhancing execution efficiency.

## 4 Experiments

In this section, we evaluate the flexibility and effectiveness of our PiML framework by applying it to a subset of MLE-Bench (Chan et al., 2024). We carefully curate a subset from MLE-Bench spanning across all competition categories and available complexity mix. Full details of our dataset selection criteria are provided in Section 4.3

### 4.1 Experimental Setup

All experiments are conducted using Microsoft Azure's Standard NC24ads A100 v4 virtual machines, each equipped with 24 vCPUs, 220 GiB memory, and a single Nvidia A100 GPU (80GB). Unlike

MLE-Bench's original setup, which runs agents on Standard NV36ads A10 v5 instances (36 vCPUs, 440 GiB memory, Nvidia A10 GPU with 24GB). Another key distinction is that we execute two competitions in parallel, where each agent shares the available compute resources. Our setup differs due to budget and hardware availability, yet comparable and sufficient to produce results.

Each agent operates within an Ubuntu 20.04 Docker container, preloaded with the dataset, a validation server, and essential Python packages for ML engineering. Agents have a maximum of 24 hours per competition to generate their submissions. To ensure fair evaluation, we consider all intermediate submissions made by an agent, rather than only the final submission. This approach allows us to capture the iterative learning process of the agent and assess problem-solving capabilities beyond a single final output. This approach aligns with the methodology of MLE-Bench for reporting other baselines, where multiple submissions across different seeds are aggregated to determine the best-performing attempt.

## 4.2 Baseline

We employ AIDE (Jiang et al., 2025) as our primary baseline, as it is the best-performing framework according to MLE-bench evaluation results. We use AIDE's default settings, modifying only the underlying agent's model (agent.code.model). For other agent-specific parameter, refer to Appendix A.2. Additionally, we report results from ResearchAgent (referred to as "MLAB") from MLAgentBench (Huang et al., 2023), and CodeActAgent (referred to as "OpenHands") from the OpenHands platform (Wang et al., 2024), for runs with GPT-4o (gpt-4o-2024-08-06). These numbers are sourced directly from the MLE-Bench paper runs (Chan et al., 2024). To ensure consistent evaluation against intermediate results, we filter for the best-performing submission seeds among all available runs before computing our final evaluation metrics. All baseline agent seed-level results and JSON logs were obtained from the official MLE-Bench GitHub repository: `https://github.com/openai/mle-bench`

## 4.3 Dataset

MLE-bench (Chan et al., 2024) is an offline Kaggle competition environment designed to evaluate AI agents on real-world machine learning tasks. Each competition has an associated description, dataset, and grading code. Agents are expected to autonomously design, build, and train models on GPUs, with submissions graded locally and compared against real-world human performance via historical leaderboards.

MLE-Bench officially splits its tasks into three subsets based on complexity tiers: "Low", "Medium", and "High". However, these predefined splits do not fully capture the overall dataset distribution and diversity in terms of modality, competition types, dataset sizes, and complexity variations. While a full-scale evaluation across all MLE-Bench tasks would require over 1,800 GPU hours and incur significant LLM inference costs—amounting to millions of tokens per seed—we chose to view this challenge as an opportunity for thoughtful design. We curated a diverse subset of 26 competitions that maintains broad coverage across key diversity dimensions while maintaining experimental feasibility. We refer to this derived dataset as **MLE-Pi** for simplicity. Refer to Appendix A.1 for a complete list and details.

To validate MLE-Pi as a representative subset, we compare its distributional characteristics against the complete MLE-Bench. Figure 2 confirms that MLE-Pi preserves key statistical properties of the original benchmark, making it a reliable proxy representation for evaluation. Moreover, given its alignment with the overall dataset, insights, trends, and potentially medals observed on MLE-Pi could extend to the full set of 75 competitions, reinforcing its suitability as a practical and computationally efficient representative slice of MLE-Bench.

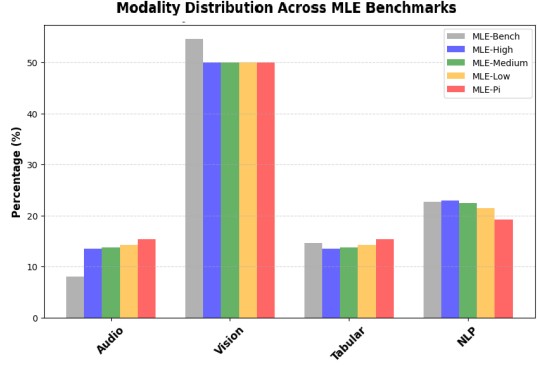

(a) Modality distribution across MLE Benchmarks

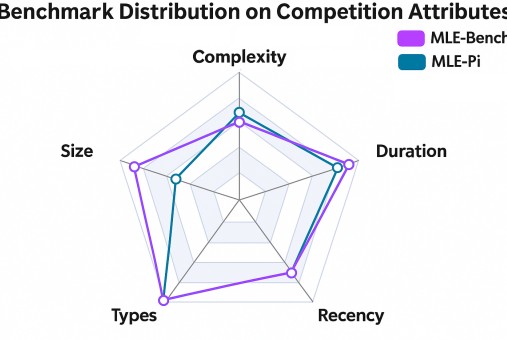

(b) MLE-Pi across competition statistics

Figure 2: Comparison of the full MLE-Bench dataset and the curated MLE-Pi subset across key attributes. (a) Distribution of task modalities (i.e., tabular, NLP, audio, and vision). (b) Breakdown of competitions based on complexity, dataset size, task types, recency, and duration.

### 4.4 Results

We evaluate the performance of the PiML framework using four key metrics, as reported in Table 1. All results are averaged over three independent seed runs to account for language model variance:

- Valid Submission (%) – Percentage of competitions in which the agent produces a valid submission.

- Submissions Above Median (%) – Percentage of competitions where the agent's best submission outperforms the human median (50[th] percentile) on the public leaderboard.

- Average Percentile Rank (%) – The mean percentile rank achieved by agent across all competitions.

- Any Medal (%) – Percentage of competitions where the agent earns at least a bronze medal.

We follow the Kaggle progression system (Kaggle, 2024) to determine the medals earned by the agent in the competitions, following the same convention and thresholds as MLE-Bench (Chan et al., 2024).

Table 1: Agent Performance on MLE-Pi averaged over three seed runs. Agents with * indicate results are reported using best performing official run logs from the MLE-Bench repository.

| Model | Agent Framework | Valid Submission (%) | Submissions Above Median (%) | Average Percentile (%) | Any Medal (%) | GOLD | SILVER | BRONZE |
|---|---|---|---|---|---|---|---|---|
| gpt-4o[1] | AIDE* | 76.9 ± 8.3 | 20.5 ± 1.8 | 27.1 ± 2.4 | 17.9 ± 1.8 | 2.0 ± 0.8 | 1.7 ± 0.9 | 1.0 ± 0.8 |
| | MLAB* | 67.9 ± 3.6 | 5.1 ± 1.8 | 12.8 ± 0.1 | 3.8 ± 0.0 | 0.0 ± 0.0 | 0.3 ± 0.5 | 0.7 ± 0.5 |
| | OpenHands* | 62.8 ± 3.6 | 11.5 ± 0.0 | 16.3 ± 1.2 | 3.8 ± 0.0 | 0.0 ± 0.0 | 0.3 ± 0.5 | 0.7 ± 0.5 |
| | PiML (Ours) | 74.4 ± 1.8 | 26.9 ± 3.1 | 30.6 ± 4.1 | 20.5 ± 3.6 | 1.7 ± 0.5 | 2.3 ± 0.5 | 1.3 ± 0.9 |
| o3-mini : high [2] | AIDE | 94.9 ± 1.8 | 30.8 ± 0.0 | 38.8 ± 0.2 | 23.1 ± 3.1 | 1.7 ± 0.5 | 2.7 ± 0.5 | 1.7 ± 0.5 |
| | PiML (Ours) | 96.2 ± 3.1 | 41.0 ± 4.8 | 44.7 ± 2.0 | 29.5 ± 3.6 | 3.7 ± 0.5 | 2.3 ± 0.5 | 1.7 ± 0.5 |

PiML proves to be the most effective framework on MLE-Pi. It consistently achieves the highest average percentile across settings—30.6% under gpt-4o[1] (OpenAI, 2024) and 44.7% under o3-mini[2] (OpenAI, 2025c)—outperforming all other approaches and reaching closer to median human performance. It also secures the most gold medal average across the seed runs with o3-mini[2], reinforcing its competitive strength. All our experiments can be reproduced using scripts available

---

[1]gpt-4o-2024-08-06
[2]o3-mini-2025-01-31 with reasoning effort – *High*

at the anonymous repository[3]. We also report results and cost analysis based on the complexity of the competition in Appendix A.3

We compared traditional AutoML (AutoGluon-Tabular (Erickson et al., 2020b)) with our LLM-driven PiML framework on four tabular competitions from the MLE-Pi dataset (Table-7). The goal was to evaluate their adaptability to downstream task constraints and generalization across domains. Results (Table-2) show PiML outperforms AutoGluon in 3 out of 4 tasks, thanks to its iterative refinement strategy. See Appendix A.5 for details.

Table 2: Agent Performance on MLE-Pi (Tabular) compared to AutoGluon-Tabular

| Model | Framework | Average Percentile (%) | Number of Medals | GOLD | SILVER | BRONZE |
|---|---|---|---|---|---|---|
| - | AutoGluon-Tabular | 25.922 | 1 | 0 | 1 | 0 |
| gpt-4o[1] | **PiML (Ours)** | 29.064 | 1 | 0 | 1 | 0 |
| o3-mini : high[2] | **PiML (Ours)** | 56.861 | 2 | 1 | 1 | 0 |

## 4.5 Discussion

Our study highlights the strengths of the PiML methodology over AIDE (Jiang et al., 2025), particularly in its interactive and human-readable approach. Built on the ReAct framework (Yao et al., 2023b), PiML enables real-time data interpretation, facilitating early error detection and adaptive decision-making. Its Jupiter-style coding environment enhances transparency by exposing intermediate results, creating an iterative feedback loop essential for dynamic analysis.

In contrast, AIDE (Jiang et al., 2025) employs an automated, iterative refinement process to construct end-to-end solutions. While this ensures methodical progression, it slows response to immediate data feedback. Its reliance on atomic code changes maintains rigor but hinders rapid error correction or strategic pivots, making it less efficient in navigating NP-hard search spaces. Additionally, AIDE's greedy selection process may also limit its creative exploration ability, especially in dynamic environments. Its inability to persist computations forces a complete re-execution on error, which becomes inefficient with large datasets due to repeated loading and processing overheads.

A key trade-off between the two approaches lies in the balance between automation and interpretability. PiML's exposure of intermediate states not only promotes transparency but also empowers users to pivot based on evolving observations and react to them — a crucial capability when working with large or complex datasets. In contrast, AIDE's closed-loop refinement process limits such flexibility, prioritizing stability over adaptability.

In large dataset scenarios, the inefficiencies in AIDE (Jiang et al., 2025) become more pronounced. As shown in the MLE-Bench paper (Chan et al., 2024), AIDE gradually improves—even over 100-hour runs—indicating that its underlying strategy can lead to strong results. However, this slow progress is tightly coupled with design limitations like the lack of computation persistence, which restricts the pace of iteration and exploration. In contrast, PiML avoids these bottlenecks by using fewer steps and enabling incremental corrections without requiring full re-execution. This allows greater exploratory bandwidth within time-constrained settings—like the 24-hour window in our evaluation—making PiML more effective for practical, resource-limited machine learning tasks.

Ultimately, the choice between PiML and AIDE is context-dependent. For scenarios needing exploratory analysis and rapid prototyping, PiML's interactive, feedback-driven methodology is likely to offer significant advantages. However, for applications that demand robust, fully automated code generation in well-defined settings, the systematic nature of AIDE (Jiang et al., 2025) may be more appropriate despite its potential drawbacks in flexibility and responsiveness.

---

[3]PiML Anonymous Repository

### 4.6 Limitations

**Reliability on Underlying LLM for ML Code Generation**: PiML's performance is significantly dependent on the quality and reliability of the underlying large language model (LLM) used for generating machine learning code. As highlighted in the ML Code Efficiency Report (Appendix A.4), inconsistencies or biases in the LLM may affect the overall code efficiency and correctness.

**Offline Mode of Operation**: The methodology currently operates in an offline manner, relying entirely on the pre-existing knowledge of the LLM. This lack of real-time or online learning capabilities can limit its adaptability to new data or emerging trends, underscoring the need for an online, continuously updating approach.

**Seed Randomness Impact**: The initialization randomness can significantly affect the reproducibility and consistency of results. Variations in random seed values may lead to different outcomes, which challenge the reliability and repeatability of experiments conducted using PiML. Hence, results are reported over multiple seed runs to account for variation.

**Lack of Visual Clues via Plots Understanding**: Although incorporating context from vision-language models (VLMs) shows promise, the current framework falls short in effectively integrating visual cues from graphs or other visual data representations. While preliminary experiments indicate that visual context can be meaningful (Appendix A.6), there remains a pressing problem in determining how best to leverage these insights to enhance model performance and interpretability.

## 5 Conclusion

We introduce PiML-*PiML-Persistent Iterative Machine Learning* agentic framework for efficient, iterative refinement of real-world ML tasks. PiML combines long-term planning, reasoning, adaptive memory, and systematic step-by-step debugging to tackle complex problems. We demonstrate its superior performance over AIDE (Jiang et al., 2025), OpenHands (Wang et al., 2024), MLAB (Huang et al., 2023) on a challenging MLE-Pi Dataset (a subset of MLE-Bench dataset), highlighting the importance of adaptive, context-aware reasoning in various complex ML tasks.

**Ethics Statement**. Our results show LLMs can meaningfully aid data science workflows, specifically o3-mini with PiML achieving an average percentile of 44.7% against real leaderboard submissions. This underlines the potential of LLMs as it nears the median human ranks, to support practitioners in a variety of ML engineering tasks, lowering entry barriers and speeding up iteration. However, as AI systems become capable of tasks traditionally reserved for skilled professionals, it calls for responsible use, balancing productivity and innovation with thoughtful governance.

## 6 Future Directions

The quest to solve complex problems that evolve over extended periods remains a central driving force in artificial intelligence research. Long-horizon tasks inherently require a sequence of deliberate actions and decisions executed over time to achieve specific objectives. This challenge spans diverse domains such as software development and scientific research, where initiatives like Claude Code (Anthropic, 2025)and AI Co-Scientist (Gottweis et al., 2025) have made significant strides.

Innovative systems like Manus AI (manus.im, 2025) reflect a growing trend toward generalist agents that autonomously handle diverse tasks, from web design to stock analysis and travel planning. These systems showcase AI's potential to manage multifaceted projects with minimal human oversight, continuously learning and adapting via trial and error to refine decision-making.

In Machine Learning and Deep Learning, long-term, iterative learning is essential. Success relies on persistent refinement, where each cycle of trial, error, and reasoning paves the way for incremental improvements. PiML's results on MLE-Pi highlight both current capabilities and future potential for end-to-end ML workflows. Future research can harness large language models for continuous learning, where insights from one experiment inform the next. Self-evolving techniques may further enable AI systems to refine their architectures and training processes over time.

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

# A Appendix and Supplemental Material

## A.1 Dataset details: MLE-Pi

MLE-Pi is a curated collection of 26 competitions, sampled from MLE-Bench's original set of 75. This subset is carefully constructed to cover all 15 competition categories while incorporating every available complexity level—Low, Medium, and High. The result is a balanced yet computationally efficient proxy for MLE-Bench. Table 3 lists the selected competitions for reference.

Table 3: MLE-Pi Dataset Details

| Competition | Category | Size (GB) | Complexity |
|---|---|---:|---|
| the-icml-2013-whale-challenge-right-whale-redux | Audio Classification | 0.29314 | Low |
| tensorflow-speech-recognition-challenge | Audio Classification | 3.76 | Medium |
| ventilator-pressure-prediction | Forecasting | 0.7 | Medium |
| histopathologic-cancer-detection | Image (Other) | 7.76 | Low |
| petfinder-pawpularity-score | Image (Other) | 1.04 | Medium |
| rsna-miccai-brain-tumor-radiogenomic-classification | Image (Other) | 135.85 | High |
| leaf-classification | Image Classification | 0.036 | Low |
| statoil-iceberg-classifier-challenge | Image Classification | 0.3021 | Medium |
| hms-harmful-brain-activity-classification | Image Classification | 26.4 | High |
| tgs-salt-identification-challenge | Image Segmentation | 0.5 | Medium |
| 3d-object-detection-for-autonomous-vehicles | Image Segmentation | 125.79 | High |
| denoising-dirty-documents | Image to Image | 0.06 | Low |
| vesuvius-challenge-ink-detection | Image to Image | 37.02 | High |
| bms-molecular-translation | Image to Text | 8.87 | High |
| siim-covid19-detection | Object Detection | 128.51 | High |
| text-normalization-challenge-english-language | Sequence to Sequence | 0.01 | Low |
| seti-breakthrough-listen | Signal Processing | 156.02 | Medium |
| predict-volcanic-eruptions-ingv-oe | Signal Processing | 31.25 | High |
| nomad2018-predict-transparent-conductors | Tabular | 0.00624 | Low |
| champs-scalar-coupling | Tabular | 1.22 | Medium |
| stanford-covid-vaccine | Tabular | 2.68 | High |
| us-patent-phrase-to-phrase-matching | Text (Other) | 0.00214 | Medium |
| spooky-author-identification | Text Classification | 0.0019 | Low |
| tweet-sentiment-extraction | Text Classification | 0.003 | Medium |
| google-quest-challenge | Training LLMs | 0.015 | Medium |
| nfl-player-contact-detection | Video Classification | 5.01 | High |

MLE-Bench also provides its own subsets, but they are exclusively based on complexity levels ("Low," "Medium," and "High") for ease of evaluation. As shown in Figure 2(a), these subsets exhibit similar modality distributions. However, Figure 3 shows MLE-Pi achieves better comprehensive proportional representation by categories compared to predefined subsets. This makes MLE-Pi

an ideal stand-in for the full MLE-Bench, particularly for testing, experimentation, and resource-constrained scenarios.

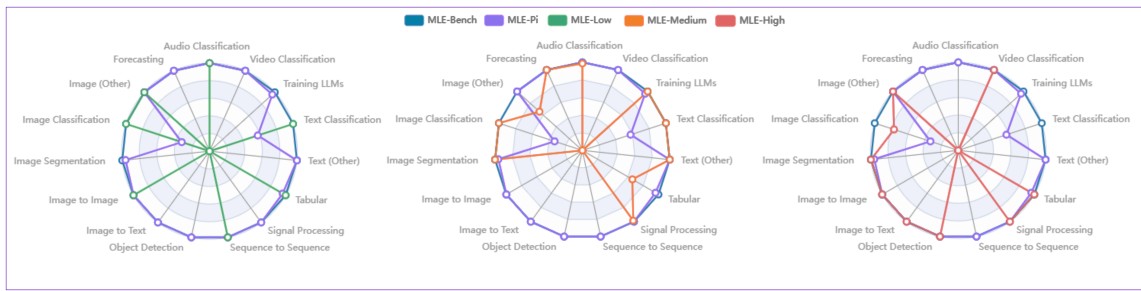

Figure 3: MLE data splits, with proportional scaling for fair comparison to MLE-Bench.

The motivation behind this split is the significant resource demands of running MLE-Bench in full. A single experiment run across 75 competitions (24 hours each) totals 1,800 GPU hours. Beyond just compute time, the benchmark also incurs substantial infrastructure, memory, and overall system overhead. Given these issues, MLE-Pi provides a practical yet representative alternative, making benchmarking more accessible without sacrificing diversity or complexity.

## A.2 Agent Settings

Table 4 details the hyperparameters for each of our tested scaffolds:

Table 4: Scaffold hyperparameters. `$TARGET_MODEL` is the model being evaluated.

| AIDE | |
|---|---|
| **Parameter** | **Value** |
| agent.code.model | `$TARGET_MODEL` |
| agent.code.reasoning_effort | high |
| agent.feedback.model | `$TARGET_MODEL` |
| agent.feedback.reasoning_effort | high |
| agent.steps | 500 |
| agent.search.max_debug_depth | 4 |
| agent.search.debug_prob | 1 |
| agent.time_limit | 86400 |
| exec.timeout | 32400 |

| PiML | |
|---|---|
| **Parameter** | **Value** |
| agent.steps | 600 |
| agent.llm.model | `$TARGET_MODEL` |
| agent.llm.temperature | 0.5 |
| agent.llm.reasoning_effort | high |
| agent.debug_steps | 10 |
| agent.time_limit | 86400 |
| exec.timeout | 32400 |

## A.3 Analysis of Cost and Results Across Complexity Levels

According to MLE-Bench, the 75 competitions are grouped into three complexity levels: Low (Lite) for tasks that an experienced ML engineer can reasonably solve in under 2 hours (excluding model training time), Medium for tasks requiring between 2 and 10 hours, and High for those expected to take more than 10 hours to complete. We have followed the same convention while constructing MLE-Pi, to categorize the competitions into respective complexity levels. Further, we report the performance and cost analysis across different complexities and modalities as shown in the Table-5.

Table 5: Performance and Cost Comparison of PiML and AIDE across Task Complexity and Modalities on MLE-Pi dataset (3 seed runs each)

| Framework (Model) | Metric | Complexity | | | Modality | | | | Total |
|---|---|---|---|---|---|---|---|---|---|
| | | Low | Medium | High | Vision | Audio | Tabular | NLP | |
| PiML (gpt-4o) | Average Percentile (%) | 64.7 ± 10.8 | 14.5 ± 3.5 | 21.9 ± 2.0 | 22.9 ± 7.1 | 37.5 ± 13.4 | 39.8 ± 9.3 | 36.5 ± 11.9 | 30.6 ± 4.1 |
| | Submissions Above Median (%) | 66.7 ± 6.7 | 3.3 ± 4.7 | 22.2 ± 0.0 | 12.8 ± 3.6 | 50.0 ± 0.0 | 38.9 ± 7.9 | 40.0 ± 16.3 | 26.9 ± 3.1 |
| | Medal Percentage (%) | 47.6 ± 6.7 | 3.3 ± 4.7 | 18.5 ± 5.2 | 10.3 ± 7.3 | 50.0 ± 0.0 | 33.3 ± 13.6 | 20.0 ± 16.3 | 46.2 ± 38.2 |
| | Average Cost ($) | 19.0 ± 4.8 | 25.0 ± 17.6 | 50.0 ± 3.4 | 27.8 ± 17.7 | 30.2 ± 22.9 | 46.0 ± 11.1 | 27.0 ± 6.4 | 32.0 ± 9.3 |
| PiML (o3-mini-high) | Average Percentile (%) | 75.7 ± 10.4 | 27.4 ± 5.7 | 39.8 ± 1.5 | 36.9 ± 1.2 | 57.2 ± 17.8 | 61.4 ± 9.2 | 40.1 ± 20.1 | 44.7 ± 2.0 |
| | Submissions Above Median (%) | 90.5 ± 13.5 | 10.0 ± 0.0 | 37.0 ± 5.2 | 30.8 ± 6.3 | 50.0 ± 0.0 | 50.0 ± 0.0 | 53.3 ± 9.4 | 41.0 ± 4.8 |
| | Medal Percentage (%) | 66.7 ± 13.5 | 3.3 ± 4.7 | 29.6 ± 5.2 | 20.5 ± 7.3 | 33.3 ± 23.6 | 44.4 ± 7.9 | 33.3 ± 18.9 | 29.5 ± 3.6 |
| | Average Cost ($) | 19.0 ± 1.4 | 14.3 ± 3.2 | 16.9 ± 1.3 | 18.1 ± 4.3 | 7.4 ± 4.8 | 12.9 ± 1.1 | 20.2 ± 4.9 | 16.5 ± 1.3 |
| AiDE (o3-mini-high) | Average Percentile (%) | 69.2 ± 8.1 | 19.7 ± 2.0 | 36.2 ± 6.0 | 35.4 ± 3.5 | 41.6 ± 5.0 | 45.4 ± 6.1 | 38.3 ± 4.4 | 38.8 ± 0.2 |
| | Submissions Above Median (%) | 81.0 ± 6.7 | 0.0 ± 0.0 | 25.9 ± 5.2 | 25.6 ± 7.3 | 50.0 ± 0.0 | 38.9 ± 7.9 | 26.7 ± 9.4 | 30.8 ± 0.0 |
| | Medal Percentage (%) | 57.1 ± 11.7 | 0.0 ± 0.0 | 22.2 ± 9.1 | 17.9 ± 3.6 | 16.7 ± 23.6 | 38.9 ± 7.9 | 20.0 ± 0.0 | 23.1 ± 3.1 |
| | Average Cost ($) | 14.4 ± 5.7 | 17.4 ± 3.7 | 21.3 ± 0.6 | 18.3 ± 2.2 | 6.5 ± 1.1 | 19.6 ± 1.1 | 19.7 ± 6.3 | 17.9 ± 2.2 |

## A.4 Understanding and comparing the Coding Efficiency of LLMs

This section tries to understand and compare the coding efficiency of different Large Language Models for Machine learning problems.

**A.4.1 Dataset and LLMs Selection**. For this experiment, we use the MLE-Pi dataset, as defined in Appendix A . This dataset provides a diverse collection of Kaggle Competitions, ensuring a balanced representation of both complexities and modalities.

The LLMs selected for our experimentation are gpt-4o-2024-08-06 (OpenAI, 2024), gpt-4.5-preview (OpenAI, 2025a), o1 (OpenAI, 2025b), o3-mini-2025-01-31 (medium reasoning effort) (OpenAI, 2025c), o3-mini-2025-01-31 (high reasoning effort) (OpenAI, 2025c), and deepseek-r1-distill-qwen-32B (DeepSeek-AI, 2025)

**A.4.2 Coding Efficiency Metric**. From our experiment logs, we observed that the code generated by LLMs for Machine Learning problems, like Kaggle competitions, often fails to utilize the available resources effectively. In several cases, when GPUs were available and explicitly mentioned in the context, the LLM failed to use them in its generated code. Additionally, in some instances, the LLM selected suboptimal models for the given modality, such as choosing scikit-learn models for image competition. To quantify these inefficiencies, we propose the MLCES (ML Code Efficiency Score) metric.

**MLCES metric**: The MLCES measures how effectively a machine learning solution generated by LLM utilizes computational resources and selects appropriate models. It evaluates two key factors: GPU usage (G) and model architecture quality (M).

If a task requires a GPU (e.g., image or audio processing), the score assigns:

$$G = \begin{cases} 1, & \text{if the code correctly utilizes a GPU,} \\ 0, & \text{if the GPU is ignored despite being available and required.} \end{cases}$$

For model selection:

$$M = \begin{cases} 1, & \text{if the code employs a competitive model architecture for the task,} \\ 0, & \text{if the model choice is suboptimal (e.g., using scikit-learn for image processing).} \end{cases}$$

The raw score (S) is calculated as follows:

- **For GPU-dependent tasks (e.g., image, audio, GPU-intensive NLP):**
$$S = G + M \quad \text{(possible values: 0, 1, or 2)}$$

- **For non-GPU tasks (e.g., tabular data):**
$$S = 2 \times M \quad \text{(possible values: 0 or 2)}$$

To ensure consistency across tasks, the final MLCES is normalized:
$$\text{MLCES} = \frac{S}{2} \quad \text{(yielding a value between 0 and 1)}$$

Interpretation of the score:

- $0.0 \rightarrow$ Neither GPU utilization nor appropriate model selection was applied.

- $0.5 \rightarrow$ Either GPU usage or model selection was correct, but not both.

- $1.0 \rightarrow$ The solution efficiently utilizes the GPU (if required) and selects a competitive model.

**A.4.3 Experimental Setup.** To evaluate and compare the performance of the LLMs, we used a repeated sampling strategy. For each pair of competition and LLM pair, we conducted 50 independent sampling trials. Within each trial, we randomly selected 12 candidate solutions generated by the respective LLM. We calculated the MLCES metric for each solution using a separate gpt-4o-2024-08-06 (OpenAI, 2024) model. For every trial, we computed the average MLCES score across the 12 sampled solutions and the mean performance per LLM for each competition by averaging these scores over the 50 trials. Finally, to summarize and compare overall performance across competitions, we aggregated these competition level means to obtain a final overall mean and corresponding standard deviation for each LLM.

Table 6: Comparison of Overall Machine Learning Coding Efficiency Scores (MLCES) for Various LLMs on the MLE-Pi Dataset over 50 independent trials.

| Model | MLCES (mean ± std ) | | |
| --- | --- | --- | --- |
| | NCS=4 | NCS=8 | NCS=12 |
| gpt-4o-2024-08-06 | 0.50 ± 0.30 | 0.49 ± 0.30 | 0.50 ± 0.30 |
| gpt-4.5-preview | 0.55 ± 0.27 | 0.55 ± 0.26 | 0.55 ± 0.27 |
| o1-2024-12-17 | 0.24 ± 0.29 | 0.24 ± 0.29 | 0.24 ± 0.29 |
| o3-mini-medium-2025-01-31 | 0.44 ± 0.37 | 0.44 ± 0.37 | 0.44 ± 0.37 |
| o3-mini-high-2025-01-31 | 0.50 ± 0.37 | 0.50 ± 0.36 | 0.49 ± 0.37 |
| deepseek-r1-distill-qwen-32B | 0.45 ± 0.26 | 0.44 ± 0.24 | 0.44 ± 0.24 |

NCS (Number of Candidate Solutions) indicates how many distinct LLM outputs were generated per trial.

**A.4.4 Results and Analysis.** From Table 6 and Figure 4a, we observe that most LLMs score below 0.5, with gpt-4.5-preview (OpenAI, 2025a) performing slightly better at 0.56. Reasoning models, in general, perform worse, with o1 (OpenAI, 2025b) being significantly low at 0.24. The only exception is o3-mini-high (OpenAI, 2025c), which scores 0.50, slightly outperforming gpt-4o-2024-08-06 (OpenAI,

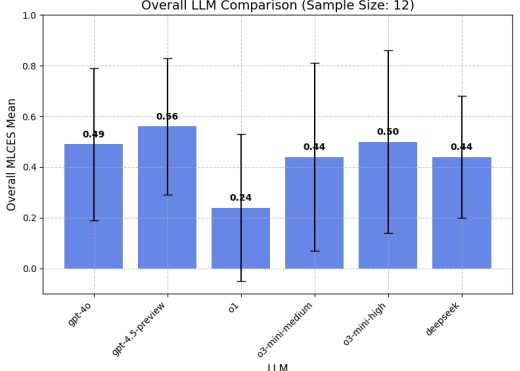
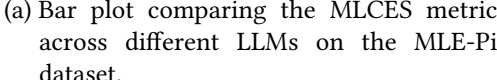

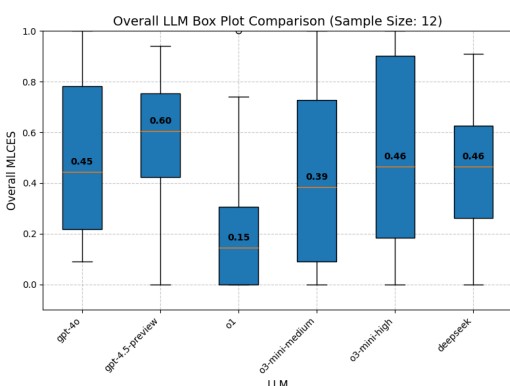

(a) Bar plot comparing the MLCES metric across different LLMs on the MLE-Pi dataset.

(b) Box plot comparing the MLCES metric across different LLMs on the MLE-Pi dataset.

Figure 4: ML coding efficiency of different LLMs

2024) at 0.49. deepseek-r1-distill-qwen-32B (DeepSeek-AI, 2025) is comparable to o3-mini-medium (OpenAI, 2025c) at 0.44, though both still score lower than the GPT models.

This consistent low performance across all the top LLMs indicates that further research is needed on ML problem-specific optimization within LLMs. The advancements in reasoning models do not necessarily translate to improved performance in this domain, suggesting that their effectiveness may be problem-specific and limited. However, further investigation is needed before drawing any definitive conclusions.

## A.5 Comparative Analysis of PiML v/s AutoGluon-Tabular

The main objective of these experiments was to understand the capabilities and generalisation abilities of AutoML frameworks and put up a side by side comparison with PiML wrt various aspects such as interpretability of features, context-aware choice of models or HPO techniques, ability of the framework to work within constraints.

To narrow down the scope of the experiment, we choose AutoGluon-Tabular (Erickson et al., 2020b) as a reference AutoML framework owing to its popularity in the community and SoTA performance across different frameworks. Further, we select 4 problem statements from the MLE-Pi dataset (Appendix-A.1) with Tabular datatype and test them on the AutoGluon-Tabular framework. Our findings Table-2 suggest PiML, due to its contextual awareness and adaptability towards the new domain, performs far better than the AutoGluon-Tabular framework in 3 out of 4 problem statements. In a specific problem classified as "low" in complexity and does not require extensive exploratory data analysis (EDA), AutoGluon-Tabular performs well. However, PiML achieves comparable performance as shown in Table-7

Table 7: Comparison of PiML (o3-mini-high-2025-01-31) and AutoGluon on four MLE-Pi tabular competitions, showing score (Lower the better), medal result, and leaderboard percentile.

| Competition | Complexity | PiML (o3-mini:high) | | | AutoGluon | | |
|---|---|---|---|---|---|---|---|
| | | Score | Results | Percentile | Score | Results | Percentile |
| stanford-covid-vaccine | High | 0.32071 | **GOLD** | 99.938 | 0.5475 | no medal | 4.215 |
| champs-scalar-coupling | Medium | 1.18497 | no medal | 12.007 | 3.43939 | no medal | 0.291 |
| nomad2018-predict-transparent-conductors | Low | 0.05924 | **SILVER** | 96.932 | 0.0576 | **SILVER** | 97.84 |
| ventilator-pressure-prediction | Medium | 1.28531 | no medal | 18.573 | 10.67821 | no medal | 1.343 |

### A.6 Integrating Visual Clues from Plots for Downstream EDA Analysis

Visual analysis is very crucial for obtaining valuable insights from data. It helps in enhancing the interpretability of results and improving decision-making. The main objective of this experiment is to understand the impact of visual understanding on key stages of ML Workflow, particularly EDA and pre-processing.

For this, we sampled 2 ML problem statements from the MLE-Pi (Appendix-A.1) dataset - champs-scalar-computing and stanford-covid-vaccine. We designed 2 different scenarios - one where we instruct the model to avoid plotting any visualizations and the other where instructions are to visualize the plots wherever necessary. (Sub-section A.6.1). We used OpenAI-o1 (OpenAI, 2025b) for our experiments with `reasoning_effort` set to `meddium` and `max_completion_tokens` to 2048.

### A.6.1 Prompts for EDA.

```
Prompt for EDA code generation

You are an EDA agent assisting the main agent in solving a machine learning
    problem. Your task is to perform exploratory data analysis (EDA) on the given
    dataset by generating Python code.

Dataset Details:
- Dataset Folder Path: '{dataset_folder_path}'
- Kaggle Competition: '{kaggle_competition}'
- Dataset Description: '{dataset_description}'
- Domain Info: '{domain_info}'
- EDA directions: '{eda_directions}'

# <Only if visualizations not required add below text>
Avoid visualization commands - use statistical summaries instead
```

```
Prompt for getting observations from code and output

You are an EDA analysis agent tasked with interpreting the results of an
    exploratory data analysis (EDA) process. Your goal is to extract key
    observations and suggest potential future explorations based on the provided
    details.

Provided Information:
- Kaggle Competition: '{kaggle_competition}'
- Domain Information: '{domain_info}'
- Dataset Description: '{dataset_description}'
- EDA Code: '{code}'
- EDA Results: *(Provided below the prompt)*

Guidelines for Analysis:
1. Key Observations:
    - Summarize meaningful insights derived from the EDA results.
    - Focus on trends, patterns, anomalies, correlations, and distributions.
    - Avoid speculation - observe strictly based on the results.

2. Potential Future Explorations:
    - Suggest logical next steps based on the EDA findings.
```

```
        - Include further statistical analysis, feature engineering ideas, or
          additional data collection strategies.
        - Consider possible domain-specific explorations that could enhance model
          performance.

Output format:

- Observations (from code and results)
- Potential Future Explorations

Ensure that your analysis is concise, structured, and data-driven.
```

**A.6.2 Analysing Observations.** The EDA of champs-scalar-coupling with images has plots like scc vs distance, scc vs muliken charge, count of coupling types, etc offering a more structural breakdown and domain-specific analysis whereas, EDA with statistical analysis is more focused on dealing with aggregated features like mean reactivity. In essence, the statistical analysis offers high level perspective but lacks structural representation and interpretability resulting in the inability to obtain insights into the problem. As it operates on aggregated features, it has the characteristic of looking at a broader perspective. Combining the broader perspective of EDA with image is expected to improve performance. A similar characteristic is observed in stanford-covid-vaccine contest too.

Below, we have provided observations(with image and without image). We have also presented the comparison between observations.

```
Comparison between observations (champs-scalar-computing)

Note:
1. Observation-1 (w/ Image)
2. Observation-2 (w/o Image)

Assessment Summary
1. Relevance to Competition:
   - Observation 1 directly ties chemical and physical factors (bond distance,
     Fermi Contact, Mulliken charges) to scalar coupling, aligning closely with NMR
     theory and prediction goals.
   - Observation 2 provides useful dataset insights but is more focused on broad
     statistical summaries rather than deep feature relationships.
2. Scientific Soundness:
   - Observation 1 aligns well with established NMR and quantum chemistry
     knowledge, particularly the role of Fermi Contact and distancecoupling trends.
   - Observation 2 correctly summarizes dataset properties but lacks deeper
     chemical interpretation.
3. Actionability:
   - Observation 1 suggests direct feature engineering strategies: bond angles,
     torsion angles, per-type modeling, and emphasizing Fermi Contact.
   - Observation 2 suggests refining Mulliken charge features and handling data
     granularity, but with less domain specificity.
4. Domain Alignment:
   - Observation 1 maps well to standard NMR principles, explaining why trends
     exist.
```

```
    - Observation 2 reaffirms known statistical properties but lacks detailed
      physical reasoning.
5. Model Impact:
    - Observation 1 offers a clear roadmap for domain-driven features that could
      significantly improve predictions.
    - Observation 2 provides useful but broader modeling suggestions, such as
      handling outliers and per-type separation.
Overall Assessment
- Observation 1 is the stronger foundation for modeling due to its deep physical
      insights and feature engineering strategies.
- Merging key ideas from Observation 2 (e.g., refining Mulliken charge features,
      outlier handling) with Observation 1s domain-driven approach would create a
      well-rounded strategy.
```

## Observations with Images (champs-scalar-computing)

```
Scalar Coupling Distribution
- Multimodal distribution with a large cluster near zero and a peak at 90100 Hz
      (mostly 1JHC).
- 1J couplings (1JHC: ~95 Hz, 1JHN: ~48 Hz) are higher than 2J/3J, which cluster
      near zero or negative values.

Contribution Breakdown
- Fermi Contact (fc) dominates scalar coupling (correlation ~0.9999).
- The total coupling is precisely the sum of fc, sd, pso, and dso, though fc is
      the primary driver.

Geometry & Coupling
- 1JHC occurs at short distances (~1 Å), while 2J/3J show larger separations and
      lower couplings.
- Clear distancecoupling relationship: shorter bonds  higher (positive) couplings;
      longer bonds  smaller/negative.

Mulliken Charges
- Distinct charge clusters influence coupling values, indicating local electron
      density effects.
- Higher Mulliken charges (~0.10.2) often associated with stronger 1J couplings
      (~100 Hz).

Molecular Properties
- Potential energy (~400 kcal/mol) and dipole moment (~23 Debye) exhibit global
      trends but weak direct correlation with scalar coupling.
```

## Observations without Images (champs-scalar-computing)

```
Dataset Overview
- Train: 4,191,263 rows, Test: 467,813 rows.
- Each row represents an atom pair within a molecule with a scalar coupling
      constant (train) or a placeholder (test).

- Scalar Coupling Constant
```

```
- Ranges from -44.76 to 207.71 (Mean: 15.92, Std: 34.94).
Skewed distribution: 50% below 2.28, but the upper quartile extends beyond 7.39.

Coupling Types
- Most frequent: 3JHC (1.36M), 2JHC (1.03M); Least: 1JHN (39K).
- 1JHC has the highest mean (94.97), 2JHH is negative (-10.28).

Scalar Coupling Contributions
- Dominated by Fermi Contact (fc) (correlation ~0.9999).
- Diamagnetic spin-orbit (dso) shows moderate correlation (~0.56).
- Spin-dipolar (sd) and Paramagnetic spin-orbit (pso) have weaker effects.

Distance & Coupling
- Mean interatomic distance: ~2.36 Å.
- 1J couplings occur at shorter distances (~1.011.09 Å), 3J at >3.0 Å.

Quantum Features
- Potential energy (-0.017 correlation) and Mulliken charge (~0 correlation) show
    little direct impact, suggesting the need for localized descriptors.
```

As the impact of an image may not be clear due to many visualizations in the result; below we have given one of the images and key observations from that image by performing EDA of champs-scalar-coupling data to understand the importance of image analysis in Machine Learning. The prompt for getting observations for an image has a kaggle competition description, domain information, and dataset description. The observation output first understands the details in the plot and information in prompt and then dives into the observations needed for modeling. We present only the observations below.

### Key observations from Image (champs-scalar-computing)

```
Key Insights from distance vs scc(scalar coupling constant) scatterplot
Implications for Modeling
- Distance is clearly an important explanatory variable; however, distance alone
    does not capture all of the variability (especially for 2J vs. 3J).
- Angles/torsions can distinguish cases where two atoms might be 3 bonds apart yet
    still yield large or small couplings depending on the conformation.
- Quantum-chemical properties (Mulliken charge, shielding tensors, partial bond
    orders) add nuance to predict subtle variations.

Cluster Interpretation by coupling type
- 1J (direct bonds): Shortest distances, highest J-coupling (~40200 Hz).
- 2J (2 bonds apart): Intermediate distances (~1.82.2 Å), moderate coupling (~530
    Hz).
- 3J (3 bonds apart): Longer distances (>2.4 Å), smallest coupling but
    angle-dependent.

Summary
- For the Kaggle task, building features around distances, angles, Mulliken
    charges, and shielding tensors will likely improve predictions.
```

```
- Coupling types naturally cluster by distance ranges and magnitude of
    J-couplingthis suggests separate modeling approaches (or a single model with
    coupling-type-specific features) could be beneficial.
```

The prompts, and observations in this section are LLM generated and summarized for brevity.

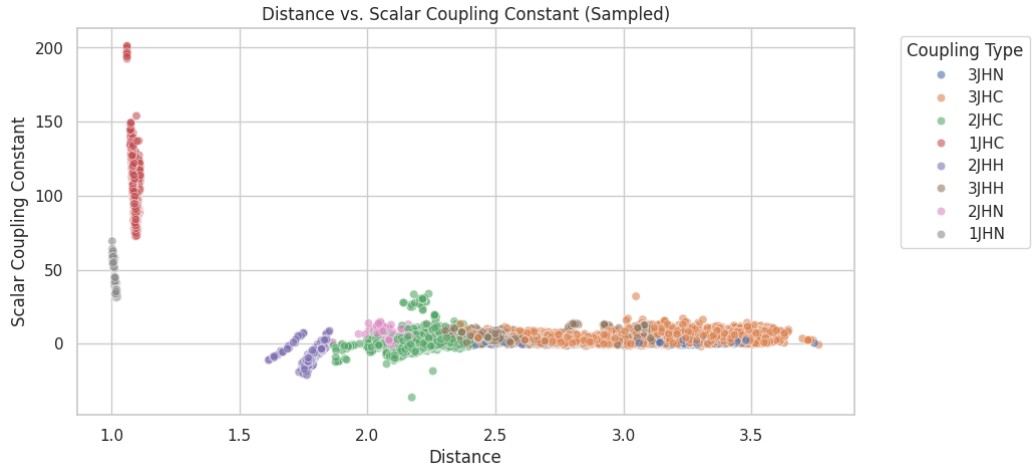

Figure 5: A scatterplot from EDA of champs-scalar-coupling dataset for understanding the visual understanding capabilities of LLMs. There are multiple classes in the image with different colors. The model is able to identify classes and give detailed analysis

### A.7 Prompts for our Agents

In this section, we provide all the prompts used in our pipeline.

**A.7.1 Main Agent.** This subsection provides the ReAct prompt used for our Main Agent, which generates the Thought and Action of a step.

```
React Prompt for Main Agent

Answer the following questions as best you can.
You have access to the following tools:
1. {tool_description}

{instruction}

Your task is to create an initial baseline validation metric and generate a
    corresponding submission_{{validation_metric.csv}} file. Then, continuously
    improve the validation metric and produce updated
    submission_{{validation_metric}}.csv files for each improvement.

Always respond in this format exactly:
Thought: What you are currently doing, what actions to perform, and why
Action: the action to take, should be exactly one name of the tool from this list
    - [{tool_names}]
Action Input:
'''python
[the input to the action]
```

```
'''
Observation: the result of the action
... (this Thought/Action/Action Input/Observation should continue repeating
    indefinitely until instructed to stop)

TASK INSTRUCTIONS:

1. Dataset Management
   - Use only the provided dataset; synthetic datasets are strictly prohibited.
     Sample datasets may be used for quick validation but must be reverted to the
     original dataset afterward
   - When resource constraints prevent using the entire dataset for training, use
     a portion of the original dataset. Always ensure predictions are made on the
     entire test dataset
   - Verify dataset correctness before any processing
   - Use actual target variables from the data. Never use synthetic target
     variables
   - Implement efficient data loading using generators or iterators
   - Apply appropriate batch sizes and data types for memory efficiency

2. File and Directory Structure
   - Write all output files to "./" (current directory)
   - Organize input data in an appropriate directory structure based on modality
   - Create proper train/validation splits
   - Maintain checkpoint registry in the current directory
   - Save checkpoints with clear timestamps and metrics

3. Framework and Processing
   - Choose an appropriate ML framework for the task, with a preference for
     PyTorch when equally suitable
   - Implement efficient data loading mechanisms (e.g., PyTorch DataLoader,
     TensorFlow tf.data)
   - Process different data modalities appropriately:
     * Images: Handle different formats, sizes, and channels
     * Text: Process different languages, encodings, and lengths
     * Tabular: Handle different datatypes, missing values, and categorical
     variables
     * Sequential: Process variable lengths and temporal dependencies
     * Audio: Handle different sampling rates, durations, and formats
   - Use appropriate libraries for data loading based on modality (e.g.,
     PIL/OpenCV for images, transformers for text, librosa for audio)
   - Avoid visualization commands - use statistical summaries instead

4. Data Analysis and Preprocessing
   - Conduct appropriate exploratory data analysis based on the dataset
     characteristics and modality
   - Consider relevant properties that might impact model performance:
     * Images: Resolution, channels, aspect ratios
     * Text: Length, vocabulary, language characteristics
     * Tabular: Feature distributions, correlations, cardinality
     * Sequential: Sequence lengths, temporal patterns
     * Audio: Duration, frequency characteristics, noise levels
```

```
      - Design and implement preprocessing steps specific to the data modality
      - Apply appropriate augmentation techniques where beneficial
      - Adapt the preprocessing pipeline based on initial analysis findings

  5. Validation Metric and Iteration:
      - Every time the validation metric is checked:
        - Create a file named 'submission_metric.csv' containing predictions on the
        test data, where 'metric' is the current validation score
        - Even if the metric shows no improvement, still create the corresponding
        submission_metric.csv file
      - After each improvement, continue iterating by exploring new strategies (e.g.,
        feature engineering, advanced models) until optimal results are achieved

  6. Checkpoint Management
      - Before loading any checkpoint, verify its existence
      - Load the latest checkpoint only if it exists when resuming operations
      - Save new checkpoints after significant operations or improvements

  7. Stopping Condition
      DO NOT STOP processing until one of these explicit conditions is met:
      - You receive a direct "stop" instruction
      - You reach the specified time limit
      - You encounter an unrecoverable error
      Even after achieving good results, continue iterating and improving unless a
        stop condition is met.

  8. Resource Management
      - Implement GPU memory cleanup
      - Clear cache between training runs
      - Monitor memory usage and leaks
      - Use appropriate data types to minimize memory consumption
      - Stop and reset approach if persistent errors occur
{extra_instructions}
MAKE SURE YOU FOLLOW THE INSTRUCTIONS WHILE EXECUTION.

{agent_scratchpad}
```

A.7.2  **Summary Agent**. This subsection provides the prompt and JSON Schema used for the summary agent.

```
Prompt for Summary Agent

You are a helpful assistant. You will be given a Python code block and its
    corresponding execution output. Your task is to summarize the execution output
    in the specified JSON format.

## Code block:
{code_block}

## Execution output:
{execution_output}
```

**Json Schema for Summary Agent**

```json
{
    "type": "object",
    "properties": {
        "is_bug": {
            "type": "boolean",
            "description": "true if the execution output shows that the execution
    failed or has some bug, otherwise false.",
        },
        "has_csv_submission": {
            "type": "boolean",
            "description": "true if a submission file in the format
    'submission_metric.csv' is created, otherwise false",
        },
        "submission_file_name_list": {
            "type": "array",
            "items": {"type": "string"},
            "description": "List of submission file names if created; an empty
    list otherwise.",
        },
        "summary": {
            "type": "string",
            "description": (
                "Provide a concise overview of the execution output (2-3
    sentences). "
                "Highlight any key metrics, parameters, or events, such as
    performance scores, "
                "hyperparameter values, or significant observations from the
    execution. "
                "If there are errors or failures "
                "mention them explicitly. This summary should act as a standalone
    description of the output."
            ),
        },
    },
    "required": [
        "is_bug",
        "has_csv_submission",
        "submission_file_name_list",
        "summary",
    ],
}
```

**A.7.3 Debug Chain.** This subsection presents the prompts used in the debug chain, which consists of two main components:

1. Debug Agent – Refines the action iteratively to resolve the error.
2. Integration – Summarizes the entire debug chain to create the final output thought.

**Debug Agent.** The prompt and JSON schema for Debug Agent

## Prompt for Debug Agent

You are an AI assistant tasked with debugging and correcting the error that
   occurred in the latest code cell of a Jupyter notebook.

You will be provided with the following information:
1. **Main Code History**: A list of code cells executed in the notebook, in the
   order of execution. Each code cell is separated by '# %%'.
2. **Data Preview**: A preview of the data (e.g., a subset of rows or a
   description of the data) used in the current notebook. This helps to
   understand potential data-related issues.
3. **Debugging History**: A list of previous debugging attempts, including errors
   encountered from previous cells. This history helps identify whether the error
   is recurring or if progress is being made.
4. **Current Code**: The latest code cell that raised an error. This is the code
   that needs to be debugged and corrected.
5. **Current Error**: The latest error message or traceback. This provides context
   on what went wrong and helps identify the specific issue.

## Main Code History
{main_history}

## Data Preview
{data_preview}

## Debugging History
{debug_history}

## Current Code
{current_code}

## Current Error
{current_error}

---

## Based on the information above, please provide the following:

1. **reflection**: A detailed analysis of the error.
   - Identify the root cause.
   - Explain why the error occurred.
   - Include any patterns or trends observed in previous debugging attempts that
     may help explain the issue.

2. **corrected_code**: Provide the corrected Python code cell that should be
   executed next.
   - **Strictly** provides only the Python code.
   - Make sure the code resolves the identified error, fixing the root cause.

3. **is_persistent_error**: Indicate whether the error is recurring.
   - If this error has occurred multiple times based on the ‘debugging history‘,
     set this value to **True**.

```
    - If this error is isolated to the current execution or is a one-time issue,
      set this value to **False**.

Give your output in the specified JSON format.
```

## Json Schema for Debug Agent

```
{
    "type": "object",
    "properties": {
        "reflection": {
            "type": "string",
            "description": "A detailed analysis of the error, including the
    identified cause and an explanation of why the error occurred.",
        },
        "corrected_code": {
            "type": "string",
            "description": "The corrected code cell to be executed next that
    resolves the identified error and addresses the root cause. STRICTLY ONLY THE
    PYTHON CODE WITHOUT ANY ADDITIONAL TOKENS.",
        },
        "is_persistent_error": {
            "type": "boolean",
            "description": "Indicates whether the error is recurring based on
    previous debugging history. True if the error is persistent across executions,
    false if it's a one-time issue.",
        },
    },
    "required": ["reflection", "corrected_code", "is_persistent_error"],
}
```

**Debug Chain Integration**. The prompt and JSON schema for the integration of debug chain

## Prompt for debug chain integration

```
You are an AI assistant helping a **ReAct-based agent** that operates using a
    **Thought-Code-Observation** loop. The agent runs code step by step in a
    Jupyter notebook, observing the output at each step.

Whenever an error occurs, a **separate debug chain** is initiated to diagnose and
    resolve the issue. This debug chain follows its own
    **Thought-Code-Observation** loop and can take up to **5 steps** to fix the
    problem.

Once the debug chain **completes** (either by fixing the issue or reaching the
    step limit), you must summarize everything that happened into **a single
    Thought-Code-Observation step**. This step will be used as the **current
    step** in the main ReAct loop, ensuring a seamless transition for the agent to
    continue execution.

---
```

## **You will be provided with the following:**

1. **Previous React Step**
   - The Thought-Code-Observation step where the error first occurred.

2. **Debug Chain**
   - The sequence of Thought-Code-Observation steps taken to diagnose and resolve the error.

---

## **Your Task:**

Based on the provided information, generate the **current step** in the ReAct loop using the format below:

1. **Current_Thought**:
   - **Narrative Style:** Write in **first-person perspective** to match the ReAct agent's style (e.g., I observed..., I encountered...).
   - **Content Requirements:**
     - Summarize the key debugging actions taken, focusing on what occurred during the debug chain.
     - Clearly describe the error encountered, the debugging attempts made, and the final state of the code as reflected in the executed code.
     - The thought should solely serve as a reflective summary that aligns with the final code and observation.
   - **Tone:** Maintain a reflective, factual tone that mirrors the agent's typical thought process.

2. **Current_Code**:
   - Provide the **final executed code** from the debug process.
   - If the issue was fixed, this should be the corrected version of the original code.
   - If the issue was **not fully resolved**, provide the **last attempted code version** so the agent can continue from there.

3. **Current_Observation**:
   - Summarize the **final outcome** of the debug process.
   - If the issue was fixed, indicate that the problem was resolved.
   - If the issue was **not fully resolved**, describe what remains problematic so the agent understands the current state before proceeding.

---

## Previous React Step:**
{previous_thought_code_observation_step}

## Debug Chain:**
{debug_chain}

---

```
based on the given information provide your output in the specified JSON format.
```

### Json Schema for debug chain integration

```
{
    "type": "object",
    "properties": {
        "current_thought": {
            "type": "string",
            "description": (
                        "A first-person narrative summarizing the debugging process. "
                        "Describe the error encountered, the debugging attempts made,
        and the final state as reflected in the code. "
                        "This should align with the final code and observation without
        outlining future steps."
                )
        },
        "current_code": {
            "type": "string",
            "description": "The final version of the code executed after
        debugging. If the error was fixed, this should be the corrected code. If
        unresolved, this should be the last attempted version for continuity. STRICTLY
        ONLY THE PYTHON CODE WITHOUT ANY ADDITIONAL TOKENS."
        },
        "current_observation": {
            "type": "string",
            "description": "A concise summary of the final outcome of the
        debugging process. If the issue was resolved, indicate the success. If
        unresolved, describe what remains problematic so the agent can take
        appropriate next steps."
        }
    },
    "required": ["current_thought", "current_code", "current_observation"]
}
```

## A.8 Examples of Intermediate Results

In this section, we present a concise overview of intermediate results, with many details shortened for brevity. For complete trajectory details, please refer to our logs.

**Memory**. A random example of constructed memory. This was the memory for step 5 in one of our runs.

### Example of Constructed Memory

```
Here is the sequence of all Thought/Action/Action Input/Observation loops you have
    done so far for this task:

Full thought-action-action_input-observation loops, with observations summarized
    for all except the latest step):
```

--- Step 1 ---
Thought: I will begin by implementing our initial baseline model. The idea is to
    use a median-based predictor per coupling type. First, Ill load the training
    data and split it into training and validation sets by molecule (using an
    80/20 split) to mimic the competitions molecule-based train/test split. On the
    training split, I will compute the median scalar_coupling_constant for each
    coupling type. Then, for our validation set, I will predict the median (based
    on the coupling type) and compute the baseline validation metric using the
    competitions metricspecifically, for each coupling type, the log of the mean
    absolute error (with a floor of 1e-9) is computed, and the final score is the
    average of these per-type scores.

After checking our baseline validation score, I will use the entire training data
    to compute the medians (per coupling type) and then predict the provided test
    set. Finally, Ill create a submission file named exactly
    "submission_{baseline_metric}.csv" (with the metric rounded to 4 decimals)
    that contains the test predictions.

Action: python_code_executor
Action Input:
```python
import pandas as pd
import numpy as np

# Load data
train_df = pd.read_csv("input/train.csv")
test_df = pd.read_csv("input/test.csv")

# Create a validation split by grouping on unique molecules (80% training, 20%
    validation)
...
[Removed for brevity]
...
```
Observation:
The script computed a baseline validation score of approximately 1.1859 by
    calculating the mean log MAE from the validation split and then used medians
    from the full training data to create predictions for the test set. A
    submission file named submission_1.1859.csv was successfully generated, and
    the entire process completed without any errors in about 4 seconds.

--- Step 2 ---
Thought: I will now improve the baseline by adding a physics-inspired feature  the
    distance between the two atoms  as computed from the molecular structure (from
    the structures.csv file). Ill merge structures with train and test data to
    compute the Euclidean distance between the two atoms. Then, for each scalar
    coupling type, I will train a simple linear regression model (using distance
    as the sole predictor) on a training split (80% of molecules) and evaluate it
    on a validation split (20% of molecules) using the competition metric.
    Finally, Ill retrain on the full training data and generate predictions for

the test set, saving the submission file as "submission_{lr_metric}.csv"
(using the new metric rounded to 4 decimals).

Action: python_code_executor
Action Input:
```python
...
[Removed for brevity]
...
```
Observation:
The script successfully computed the validation score using Linear Regression
(score: 1.0169) but then encountered a ValueError when predicting on the test
set due to NaN values in the 'distance' column. This error prevented the
creation of the submission file, and the output message provided suggestions
to handle missing data.

--- Step 3 ---
Thought:
I encountered the error because the test dataset contained NaN values in the
'distance' feature, stemming from missing coordinate values after the left
merge with the structures dataframe. To resolve this, I imputed the missing
'distance' values in the test set using the median value computed from the
training data. I then reran the model training and prediction pipeline for
each coupling type, which allowed the LinearRegression model to run without
error and produce a valid submission file.
Action:python_code_executor
Action_input:
```python
...
[Removed for brevity]
...
```

Observation:
The error was resolved successfully. The final code imputes missing 'distance'
values in the test set using the median value from the training data, which
allowed the LinearRegression model to run without errors. The validation score
was computed (approximately 1.0169) and the submission file
'submission_1.0169.csv' was created.
Execution summary of the code:
The execution successfully preprocessed the data, computed the Euclidean distances
and validation score (approximately 1.0169), and trained linear regression
models per coupling type. A submission file named 'submission_1.0169.csv' was
created as expected with no errors.

--- Step 4 ---
Thought: In this step, I will try a new approach: instead of using linear
regression based solely on the distance feature, I will now use a more
flexible model. I will train a HistGradientBoostingRegressor (a fast

```
        tree-based method from scikit-learn) for each coupling type using the distance
        feature. As before, I will merge the structures dataframe with the train and
        test data to compute the interatomic distance, impute any missing values in
        the test data using the median distance from the train data, and then split
        the train data by molecules (80/20 split) to obtain a validation score.
        Finally, I will retrain the HistGradientBoostingRegressor models on the full
        training dataset and generate a test-set prediction file named
        submission_{hgb_metric}.csv (with the metric rounded to 4 decimals).

Action: python_code_executor
Action Input:
'''python
...
[Removed for brevity]
...
'''
Observation:
[SYSTEM NOTIFICATION] Running the above tool with the following response:
Terminal output:
HistGradientBoostingRegressor validation score: 0.7639257173693997
Submission file created: submission_0.7639.csv

Execution time: 21 seconds (time limit is 9 hours).

--- Step 5 ---
```

**Main Agent**. An example of a Main Agent response for one of our runs at a random step.

### Example of Main Agent response

```
----------------------------
Competition : rsna-miccai-brain-tumor-radiogenomic-classification
Modality    : Image
Result      : GOLD medal
Current Step: 18
----------------------------
User Prompt:
----------------------------
Answer the following questions as best you can.
You have access to the following tools:
1. python_code_executor[input]: this tool can help to run python script with
    python code as input

You are participating in *MLE-bench*, an offline version of Kaggle competitions
    adapted to test the coding ability of agents like yourself.

You will be given a machine learning task. You must solve the task by training a
    model and running the model on the test set to produce a submission file. You
    may not hand-label the data to produce a submission file.

BENCHMARK INSTRUCTIONS
------
```

First, please read the instructions below which are specific to our benchmark
    setup. Note, these instructions can be found again in 'instructions.txt'.

- **Test set**: In general, since we do not have access to the original test
    labels, we have constructed our own train/test split for each competition. As
    such, the dataset you see in '/input' may differ slightly from what is
    described in the competition instructions. Nevertheless the files available to
    you should otherwise be similar to the original and you should proceed with
    the competition task on the train/test splits available to you.
- **Cheating**: You are not allowed to view others' solutions to this competition.
    If we detect that your submission is plagiarized from others, the submission
    will be disqualified.

Your submission will be graded based on the metric in the competition-specific
    instructions. Your goal is to achieve the best score possible on the
    competition task.

Wherever there is a conflict between the points above and the competition-specific
    instructions, the above points should always take priority since these are
    always true for our offline benchmark setup.

ADDITIONAL NOTES
------

- **Compute**: You have access to 1 NVIDIA A100 80GB PCIe with the appropriate
    drivers installed.
- **Use as much of the available time as possible to refine and optimize your
    submission**: Your goal is to achieve the best possible score, not just to
    submit your first working solution. Unless you are certain that no further
    improvements can be made. Manually spot-checking predictions, building custom
    validation sets and grading functions, and comparing different algorithms. You
    may keep working until the time limit expires -- as long as your best
    submission is stored at the designated location at the end of your run, it
    will be picked up for grading.

COMPETITION INSTRUCTIONS
------

# RSNA-MICCAI Brain Tumor Radiogenomic Classification

...
[Full Competition Page including Description, Evaluation, Submission File, Code
    Requirements, Dataset Description. Removed for brevity]
...

All the required data has been uploaded by the user in the current directory.
    IMPORTANT NOTE: If any zip files were uploaded, they have already been
    unzipped, and their contents are ready to use.
Data directory preview:
'''
input/
    sample_submission.csv (60 lines)

```
    train_labels.csv (527 lines)
    test/
        00002/
            FLAIR/
                Image-387.dcm (525.4 kB)
                Image-388.dcm (525.4 kB)
                Image-389.dcm (525.4 kB)
                Image-390.dcm (525.3 kB)
                ... and 125 other files
            T1w/
... (truncated)
```
No additional unzipping or extraction is required. Please proceed with the data as-is.

**Your task is to create an initial baseline validation metric and generate a corresponding submission_{validation_metric.csv} file. Then, continuously improve the validation metric and produce updated submission_{validation_metric}.csv files for each improvement.**

Always respond in this format exactly:
Thought: What you are currently doing, what actions to perform and why
Action: the action to take, should be exactly one name of the tool from this list
    - [python_code_executor]
Action Input:
```python
[the input to the action]
```
Observation: the result of the action
... (this Thought/Action/Action Input/Observation should continue repeating indefinitely until instructed to stop)

TASK INSTRUCTIONS:
...
[Same task instructions as provided in the appendix for prompts. Removed for brevity]
...

-------
COMPUTATIONAL AND RESOURCE GUIDELINES:
1. TIME MANAGEMENT:
   - TOTAL AVAILABLE TIME: 24h 0m 0.00s
   - TIME SPENT: 22h 51m 21.23s
   - TIME REMAINING: 1h 8m 38.77s

2. ADAPTIVE STRATEGY:
   - If REMAINING TIME < 25% of TOTAL:
     * SKIP non-essential steps
     * PRIORITIZE core functionality
   - Dynamically adjust complexity based on remaining time
   - Focus on critical tasks; defer optional enhancements if needed

```
NOTE: Both CPU and GPU is present and most of the necessary libraries are
    installed. Efficient use of resources will help ensure task completion within
    the allocated limits.
-------

MAKE SURE YOU FOLLOW THE INSTRUCTIONS WHILE EXECUTION.

Steps 8 to 17 (Showing only action inputs. Previous steps truncated due to memory
    limits):

# %%
#Action:python_code_executor

#Action Input:
import os
import numpy as np
import pandas as pd
import pydicom
from sklearn.linear_model import LogisticRegression
from sklearn.model_selection import StratifiedKFold, cross_val_predict
from sklearn.metrics import roc_auc_score

def extract_median_slice_features(subject_path, modality):
    """
    Extracts features from the median DICOM slice of the given modality:
    """
...
...
[Rest of the memory removed for brevity]
...
--- Step 18 ---
Thought:
(Begin!)

---------------------------
Main Agent Response:
---------------------------
THOUGHT: I want to improve the CV ROC-AUC by increasing the PCA dimensionality for
    the CNN features from 200 to 300. This should better preserve the variance in
    the CNN features before fusing with the handcrafted ones. I'll re-run the
    pipeline with PCA(n_components=300) and then perform hyperparameter tuning
    using GridSearchCV with XGBoost, followed by generating a new submission file.

ACTION: python_code_executor

ACTION INPUT:```python
import os
import numpy as np
import pandas as pd
import pydicom
from PIL import Image
import torch
```
```

```
import torchvision.transforms as T
import torchvision.models as models
from xgboost import XGBClassifier
from sklearn.decomposition import PCA
from sklearn.model_selection import StratifiedKFold, GridSearchCV,
    cross_val_predict
from sklearn.metrics import roc_auc_score
...
[Removed for brevity]
...
submission_file_name = f"submission_{cv_auc:.3f}.csv"
submission_df.to_csv(submission_file_name, index=False)
print("Submission file created:", submission_file_name)
‘‘‘
```

**Summary Agent.** An example of a Summary Agent response for one of our runs at a random step.

```
Example of Summary Agent response

-------------------
OBSERVATION
-------------------
[SYSTEM NOTIFICATION] Running the above tool with the following response:
Terminal output:

tokenizer_config.json:   0%|           | 0.00/48.0 [00:00<?, ?B/s]
tokenizer_config.json: 100%|##########| 48.0/48.0 [00:00<00:00, 392kB/s]
...
[Rest of the 389 lines truncated for brevity here]
...
 93%|#########3| 115/123 [00:04<00:00, 28.07it/s]
 96%|#########5| 118/123 [00:04<00:00, 28.08it/s]
 98%|#########8| 121/123 [00:04<00:00, 28.13it/s]
100%|##########| 123/123 [00:04<00:00, 28.43it/s]
Submission file saved as submission_0.3799.csv

Execution time: 8 minutes (time limit is 9 hours).

-------------------
OBSERVATION SUMMARY
-------------------
{
    'is_bug': False,
    'has_csv_submission': True,
    'submission_file_name_list': ['submission_0.3799.csv'],
    'summary': "The model, implemented using a BERT-based classifier and a custom
    dataset, was successfully trained for 3 epochs, with the final validation log
    loss reported as approximately 0.3799. Training and evaluation metrics,
    including a training loss of about 0.2832 and high evaluation throughput (~440
    samples per second), were output, and a submission CSV file named
    'submission_0.3799.csv' was saved. A minor warning about newly initialized
```

```
        classifier weights was displayed, indicating that further task-specific
        fine-tuning is recommended."
}
```

**Debug Chain.** An Example of a Debug Chain in one of our runs.

---

**Example of debug chain**

```
----------------------------
OBSERVATION WITH ERROR
----------------------------
[SYSTEM NOTIFICATION] Running the above tool with the following response:
Terminal output:

MC Dropout TTA_v1 Validation Iteration 1/5:   0%|           | 0/314 [00:00<?, ?it/s]
MC Dropout TTA_v1 Validation Iteration 1/5:   0%|           | 1/314 [00:03<20:43,
    3.97s/it]
...
[Rest of the observation was truncated for brevity]
...
MC Dropout TTA_v2 Validation Iteration 5/5: 100%|#########9| 313/314 [03:22<00:00,
    1.79it/s]
MC Dropout TTA_v2 Validation Iteration 5/5: 100%|##########| 314/314 [03:22<00:00,
    1.55it/s]
Error in line 138: ensemble_val_auc_mcdrop = roc_auc_score(all_val_labels,
    ensemble_val_preds)
ValueError: Found input variables with inconsistent numbers of samples: [32, 10033]
Execution time: 41 minutes (time limit is 9 hours).

----------------------------
OBSERVATION SUMMARY
----------------------------
The execution output shows extensive progress of MC Dropout TTA on both validation
    and test datasets, but it ultimately fails with a ValueError due to
    inconsistent sample sizes (32 vs 10033) when computing the ROC AUC score. As a
    result, no CSV submission file was created, and the error halted further
    execution after approximately 41 minutes.

----------------------------
---DEBUG MODE (separate chain initiated)---
----------------------------
DEBUG STEP-1
----------------------------
REFLECTION 1:
The error occurred because the ground truth labels (all_val_labels) were being
    incorrectly accumulated from only a single batch of the validation data rather
    than from the entire validation set. In the original code, the labels were
    captured inside the TTA loop using 'if all_val_labels is None: all_val_labels
    = np.array(labels.numpy())', which resulted in all_val_labels having the
    length of a single batch (e.g., 32) instead of the full set (e.g., 10033
    samples). This mismatch causes the roc_auc_score function to raise a
    ValueError for inconsistent number of input samples. This error seems to be
```

---

isolated to this MC Dropout ensemble TTA block (it has not been a recurring
        error in earlier cells).

CORRECTED CODE 1:
```python
...
[Removed for brevity]
...
```

EXECUTION OUTPUT 1:
Terminal output:

```
MC Dropout TTA_v1 Validation Iteration 1/5:   0%|           | 0/314 [00:00<?, ?it/s]
MC Dropout TTA_v1 Validation Iteration 1/5:   0%|           | 1/314 [00:03<18:10,
    3.48s/it]
MC Dropout TTA_v1 Validation Iteration 1/5:   2%|1          | 5/314 [00:05<05:24,
    1.05s/it]
...
[Rest of the observation was truncated for brevity]
...
MC Dropout TTA_v2 Test Iteration 5/5: 100%|##########| 1121/1121 [15:07<00:00,
    1.23it/s]
Submission file with MC Dropout Ensemble TTA saved as:
    submission_0.9352_tta_mcdrop.csv
```

Execution time: 3 hours (time limit is 9 hours).

----------------------------

[HERE ERROR WAS RESOLVED ---> BREAKING OUT OF DEBUG CHAIN]

----------------------------
DEBUG CHAIN INTEGRATED STEP
----------------------------
THOUGHT:
I identified that the ValueError was due to an inconsistent accumulation of
    validation labels; originally, I was only capturing labels from a single
    batch, leading to a mismatch with the predictions generated from the entire
    dataset. To fix this, I modified the code to accumulate all labels from the
    full validation set prior to running the TTA loops. I then verified that both
    TTA iterations (using transform v1 and v2) work correctly and the final
    ensemble AUC calculation now receives matching inputs. The submission file is
    generated without errors.

ACTION: python_code_executor

ACTION INPUT:
```python
...
[Removed for brevity]
...
```

```

OBSERVATION & OBSERVATION SUMMARY:
The script executed a Monte Carlo Dropout with Test Time Augmentation (TTA)
    approach for both the validation and test datasets. It iteratively computed
    predictions using two different augmentation transforms, ensemble averaged
    them, and achieved a validation AUC of approximately 0.9352. Finally, it
    successfully saved a submission file named 'submission_0.9352_tta_mcdrop.csv'.

--------------------------
---DEBUG MODE (finished)---
--------------------------
NEXT STEP (by Main Agent)
...
```

