# OpenReview forum: "PiML: Automated Machine Learning Workflow Optimization using LLM Agents"
_automl.cc/AutoML/2025/Methods_Track — AutoML 2025 Methods Track_

### Official Review · Reviewer_iPdg · 2025-04-29

**Comments To Authors:**

## **Summary**

The authors propose **PiML**, a multi-agent framework designed to autonomously solve real-world machine learning tasks. PiML demonstrates strong performance on the MLE-Bench benchmark across multiple metrics. A key contribution is its emphasis on autonomy—going beyond existing AutoML tools—by integrating components for debugging, reasoning, and adaptive task management.

---

## **Strengths**

- **Addresses key limitations of prior work**, by introducing features such as adaptive memory management and robust error handling.
- **Subset selection of MLE-Bench** is well-justified through statistical comparisons.
- Provides a **discussion of when PiML outperforms AIDE** and when the opposite holds.
- The paper is **well written and easy to follow**.

---

## **Weaknesses**


- **No uncertainty quantification**: Although the limitations section acknowledges that initialization randomness and seed variability can significantly affect results, the authors neither conduct multiple runs nor explain why this analysis was omitted. This raises concerns about the **reproducibility and robustness** of the reported performance.
- **Missing broader impact statement**: There is no discussion of potential societal implications or ethical considerations of the proposed system.

---

## **Minor Issues**

- **Algorithm pseudocode**: The algorithm section is hard to follow due to heavy use of indices and dense code formatting. A higher-level pseudocode abstraction would improve readability.
- **Table 1 – Valid Submission Rates**: PiML is visually highlighted, even though its valid submission rate is the same as AIDE’s (potentially due to rounding ?).
- **Comparison to AIDE**: A concise summary or table outlining the main conceptual and architectural differences between PiML and AIDE would be helpful for clarity.

---

## **Overall Evaluation**

The idea of building AutoML tools that leverage LLM-driven agents is promising, as well as the presented results. However, the **lack of repeated runs and uncertainty analysis**, limits the generalizability and credibility of the findings.

**Review Confidence:**

2

**Review Rating:**

7

---

### Official Review · Reviewer_qSfa · 2025-04-30

**Comments To Authors:**

&nbsp;

**__SUMMARY__**

&nbsp;

The authors introduce PiML, an agentic AutoML framework. The paper is well-written and the results appear of interest to the conference. As such, I recommend acceptance with the following points the authors may wish to consider.

&nbsp;

**__MAJOR POINTS__**

&nbsp;

1. I am not able to view certain portions of the codebase e.g. the `README` or `piml_step_submission.py` files presumably due to an issue on the part of Anonymous GitHub. I will wait to see if this problem resolves itself. For the portions that are viewable I would recommend running the code through GPT/Claude to add documentation.

2. In terms of related work on solving long complex tasks using agentic systems it would be worth mentioning [2,3] which also tackle AutoML tasks and [4] which is a general framework for training agents on complex, multiturn environments using tools.

&nbsp;

**__MINOR POINTS__**

&nbsp;

1. Reference [1] was accepted at ICLR 2024. Similarly Wei et al. was accepted at NeurIPS 2022. It would be great if missing journal and conference references were included in the camera-ready version of the paper.

2. There are some missing capitalizations in the references e.g. "AI" and "LLMs".

3. It would be great if parenthetical e.g. (Eriksson et al. 2020) and narrative citations e.g. Eriksoson et al. (2020) could be used appropriately.

4. Typo Line 29, "Tree of Thoughts".

5. The formatting of Section 2.2 could be improved e.g. if the authors intend to use mathematical notation punctuation should be added at the end of equations.

6. In Section 2.3 why is the first step indexed by k?

7. Could the authors explain why 10 appears in the case for comprehensive memory in Section 2.4.

8. Line 300, sentence probably needs to be revised.

9. Line 482, "with its results being clearer extension in performance to actual benchmark.". What do the authors mean by this?

10. Missing full stop in the caption of Figure 3.

11. Missing full stop in the caption of Table 4. This trend is apparent in quite a few figures and tables. Most style guides use consistent formatting for tables and figures e.g. full stop or no full stop but not a mix of both.

12. In Table 4, it would be great if the number of independent trials (50) was provided in the caption directly.

13. Line 557, typo, "enhancing".

14. Line 145/146, it would be worth revising this sentence.

15. What do the authors think is the reason for the low number of valid submissions for PiML relative to the other baselines in Table 1? Despite this, PiML achieves strong performance for the submissions it produces that are valid.

16. Section A.5.2 could be rewritten for enhanced clarity.

&nbsp;

**__REFERENCES__**

&nbsp;

[1] Chen, X., Lin, M., Schärli, N. and Zhou, D., [Teaching Large Language Models to Self-Debug.](https://openreview.net/forum?id=KuPixIqPiq) In The Twelfth International Conference on Learning Representations. 2024.

[2] Trirat, P., Jeong, W. and Hwang, S.J., 2024. [AutoML-agent: A multi-agent LLM framework for full-pipeline AutoML](https://arxiv.org/abs/2410.02958). arXiv preprint arXiv:2410.02958.

[3] Kon, P.T.J., Liu, J., Ding, Q., Qiu, Y., Yang, Z., Huang, Y., Srinivasa, J., Lee, M., Chowdhury, M. and Chen, A., 2025. [Curie: Toward rigorous and automated scientific experimentation with AI agents.](https://arxiv.org/abs/2502.16069) arXiv preprint arXiv:2502.16069.

[4] Narayanan, S., Braza, J.D., Griffiths, R.R., Ponnapati, M., Bou, A., Laurent, J., Kabeli, O., Wellawatte, G., Cox, S., Rodriques, S.G. and White, A.D., 2024. [Aviary: training language agents on challenging scientific tasks.](https://arxiv.org/abs/2412.21154) arXiv preprint arXiv:2412.21154.

&nbsp;

**Review Confidence:**

4

**Review Rating:**

8

---

### Official Review · Reviewer_TLcW · 2025-05-02

**Comments To Authors:**

**Paper overview**

This paper introduces PiML, a novel agent-based framework for automating the end-to-end machine learning workflow using large language models (LLMs). It is designed to handle real-world ML challenges, such as Kaggle competitions, by iteratively performing exploratory data analysis, feature engineering, modeling, and hyperparameter tuning. The system integrates a reasoning-enabled “Main Agent”, a “Summary Agent” for observation feedback, and a structured “Debug Chain” for iterative error correction. The authors also introduce an adaptive memory mechanism and define a new metric, the ML Code Efficiency Score, to evaluate how effectively a generated ML solution utilizes computational resources and selects appropriate models.

PiML is evaluated on MLE-Pi, a 26-task subset of MLE-Bench chosen by the authors, with performance benchmarks against established agentic AutoML systems such as AIDE, OpenHands, MLAB and AutoGluon-Tabular. Results show that PiML achieves higher average percentile ranks and more consistent medal attainment.

Overall, despite the core ideas (agent loop, memory, debug) build on known LLM agentic paradigms (hence the paper has limited originality in these core ideas), I believe that their combination results in an interesting pipeline being a step forward in LLM-driven AutoML systems. Limitations such as reliance on offline LLM reasoning, dependency on the LLM model quality for code generation, and lack of visual tools are acknowledged and considered for future work. The authors provide an anonymous GitHub for reproducibility.

The paper is reasonably well written, though could benefit from polishing. Also, the amount of space used for the final sections (“Limitations, “Related work”, and “Future directions”) is quite unbalanced with the space dedicated to results and discussion. I strongly believe the paper would benefit from more results shown and discussed in the main paper rather than the Appendix. Please find some comments below.

**Comments**

-	I am not sure about the choice of the method’s acronym – I believe it is not straightforward. In “PiML: Automated Machine Learning Workflow Optimization using LLM Agents” there is no real word reminding to the Pi in the acronym, while some aspects like LLMs and automation are not reflected at all in the chosen acronym. Also, PiML is already used in the literature (https://arxiv.org/pdf/2305.04214) to denote a package for a Python toolbox for interpretable machine learning which is completely detached from OpenAI. Therefore, I suggest revising it.
-	In the central block of Figure 1, it is not clear that the flowchart starts from the thought and action (code) generated by the Main Agent after being prompted, which makes it confusing.
-	The used 26-task subset of MLE-Bench defined by the authors, MLE-Pi, is not defined in the first instance it is used (line 52).
-	Most of the remarks in the Discussion Section (for example that “PiML truly shines when dealing with large dataset”) are not supported by results in the main paper and the reader is not even pointed to an Appendix section where results supporting this could potentially be shown. The paper would benefit from more results shown in the main paper rather than the Appendix and sections like “Related work” to be quickly mentioned in the Intro and moved to the Appendix instead.
-	Also, I find the choice of having a “Related work” section at the end of the paper, right before the Conclusions, quite inconvenient for a reader that is not fully familiar with the topic.
-	While PiML demonstrates promising results in automating end-to-end ML workflows, the paper does not provide a quantitative comparison of its computational cost relative to traditional AutoML frameworks. Given the reliance on iterative LLM-based reasoning, code generation, and debugging, one can reasonably expect significantly higher resource usage. Although the authors acknowledge compute constraints and introduce a smaller MLE-Pi benchmark for feasibility, a clearer analysis or discussion of runtime efficiency, energy consumption, or scalability trade-offs are missing.
-	In Appendix A.4, the authors state that in the problem where AutoGluon performs better than PiML, the latter achieves comparable performance. On the contrary, what is the performance of AutoGluon in the cases where PiML wins? This cannot be read from Table 5.

**Minor comments**

-	Caption of Figure 1: “EDA” acronym for exploratory data analysis not defined
-	Bad resolution of Figure 1.
-	The trajectory symbol notation is not consistent in Section 2.2, with bold and standard font mixed.
-	Line 158: Missing blank space after period, “submission.This”
-	The meaning of the star in Table 1 is not defined. Does this mean that data are extracted from available results?
-	Line 498: Extra space before the first comma.

**Review Confidence:**

3

**Review Rating:**

6

---

### Meta-Review · Area_Chair_sTSc · 2025-05-11

**Recommendation:** Accept
**Confidence:** 5

**Metareview:**

An agentic AI framework for approaching machine learning tasks specified in the form of challenges (e.g., those from Kaggle) is introduced. The methodology is presented with enough detail, and an experimental evaluation is performed on relevant benchmarks.  Overall, the paper proposes a novel solution to a very relevant problem (although some components are straightforward), and reviewers are mostly positive about this submission. An important concern raised by reviewers is that there is no comprehensive evaluation of uncertainty, which I think is critical.  After discussion, I think the paper could be accepted for publication, provided the authors moderate their claims on the conclusiveness of their findings and acknowledge the lack of uncertainty estimation in the paper (this recommendation was only possible because of the positive reproducibility review). Also, authors should discuss the potential societal-ethical impact of their work.